# Genetic and Epigenetic Regulation of the Smoothened Gene (SMO) in Cancer Cells

**DOI:** 10.3390/cancers12082219

**Published:** 2020-08-08

**Authors:** Hong Lou, Hongchuan Li, Andrew R. Huehn, Nadya I. Tarasova, Bahara Saleh, Stephen K. Anderson, Michael Dean

**Affiliations:** 1Laboratory of Translational Genomics, Division of Cancer Epidemiology and Genetics, Leidos Biomedical Research, Inc., National Laboratory for Cancer Research, Gaithersburg, MD 20892, USA; louho@mail.nih.gov; 2Basic Science Program, Frederick National Laboratory for Cancer Research, Frederick, MD 21702, USA; lihongchu@mail.nih.gov; 3Laboratory of Cancer Immunometabolism, Center for Cancer Research, National Cancer Institute, Frederick, MD 21702, USA; andrew.huehn@yale.edu (A.R.H.); tarasovn@mail.nih.gov (N.I.T.); baharasaleh93@gmail.com (B.S.); 4Department of Molecular Biophysics and Biochemistry, Yale University, New Haven, CT 06510, USA; 5Laboratory of Translational Genomics, Division of Cancer Epidemiology and Genetics, National Cancer Institute, Gaithersburg, MD 20892, USA

**Keywords:** hedgehog pathway, smoothened, promoter, methylation, transcription factors

## Abstract

(1) Background: The hedgehog (HH) signaling pathway is a key regulator of embryonic patterning, tissue regeneration, stem cell renewal, and cancer growth. The smoothened (SMO) protein regulates the HH signaling pathway and has demonstrated oncogenic activity. (2) Methods: To clarify the role of the HH signaling pathway in tumorigenesis, the expression profile of key HH signaling molecules, including *SMO*, *PTCH1*, *GLI1*, *GLI2*, and *GLI3*, were determined in 33 cancer cell lines and normal prostate cells and tissues. We performed a computational analysis of the upstream region of the *SMO* gene to identify the regulatory elements. (3) Results: Three potential CpG islands and several putative *SMO* promoter elements were identified. Luciferase reporter assays mapped key *SMO* promoter elements, and functional binding sites for SP1, AP1, CREB, and AP-2α transcription factors in the core *SMO* promoter region were confirmed. A hypermethylated *SMO* promoter was identified in several cancer cell lines suggesting an important role for epigenetic silencing of *SMO* expression in certain cancer cells. (4) Discussion: These results have important implications for our understanding of regulatory mechanisms controlling HH pathway activity and the molecular basis of *SMO* gene function. Moreover, this study may prove valuable for future research aimed at producing therapeutic downregulation of *SMO* expression in cancer cells.

## 1. Introduction

The hedgehog (HH) pathway is one of the key signaling pathways regulating embryonic patterning, tissue regeneration, stem cell renewal, and cancer growth [1,2,3,4]. Canonical HH signaling is triggered by the binding of HH ligand to its receptor PTCH1, resulting in the release of PTCH1-mediated repression of the seven-transmembrane protein smoothened (SMO). Activation of SMO ultimately triggers GLI-dependent expression of downstream target genes through a complex network of post-translational processes and translocations [5]. In the absence of HH ligands, PTCH inhibits SMO, GLI2 and GLI3, which are phosphorylated and undergo partial proteasome degradation, resulting in repressive forms of GLI2 and GLI3 (GLI2/3 R), which are translocated into the nucleus where they inhibit the transcription of HH target genes [6,7]. HH pathway activation amplifies the signal by increasing GLI1 levels, and in contrast, potentiates negative regulators such as PTCH1 and HH interacting protein (HHIP) [8,9]. The positive and negative feedback loops ensure that the activity of HH signaling is kept within an optimal range. Constitutive activation of the HH pathway has been observed in various types of malignancies caused either by mutations in the pathway, such as PTCH1 loss-of function or SMO activation in basal cell carcinoma, or through HH overexpression, as observed in small-cell lung cancer, glioma, endometrial carcinoma, digestive tract tumors, pancreas, and prostate [10,11,12,13,14,15,16].

The switch between active and inactive states of the HH pathway involves rapid translocation of SMO. The SMO protein is the key positive regulator of the HH pathway, and GLI family proteins play a critical role in the regulation of HH signaling pathway activity. Despite a strong link between SMO expression, HH pathway activity, and cancer development, the basis for SMO gene regulation has not been well characterized. Therefore, an investigation of the mechanisms controlling the expression of SMO and additional HH pathway genes may provide valuable insight into HH signaling alterations associated with cancer development. SMO also is the major target for pharmaceutical agents that modulate HH pathway activity [17,18,19], such as vismodegib [20] and sonidegib [21]. We previously studied SMO peptides and found that specific lipopeptides can serve as effective inhibitors [22,23].

DNA methylation of HH pathway genes is a potential regulatory mechanism in the progression of cancers. Several epigenetic factors that act on the HH signaling pathways have been associated with cancer initiation and progression [10,24]. It was reported that distinct subgroups of cancers have an exceptionally high frequency of cancer-specific CpG island hypermethylation [25,26]. Methylation has been studied as a clinical biomarker for the diagnosis and prognostic evaluation of various cancers, especially in breast cancer [27]. Recently, SMO methylation was used as a biomarker for the occurrence and development of breast cancer [28].

In the current study, we have developed a qRT-PCR method to accurately determine the expression levels of SMO, PTCH1, GLI1, GLI2, and GLI3 in a panel of cancer cell lines. Different SMO expression patterns in the cancer cell lines led us to characterize SMO gene regulatory elements. The SMO 5′-flanking region and Exon 1 were analyzed in silico, revealing that the region surrounding the SMO transcriptional start site (TSS) has an extremely high GC content (70%+) that prevents its PCR amplification by traditional methods. We used a touchdown PCR method to amplify SMO promoter fragments and determined their promoter activity using a dual luciferase assay. EMSA analysis identified binding sites for the transcription factors, SP1, AP1, CREB, and AP-2α, which likely play an important role in SMO transcriptional activity in cancer cells. To gain insight into the epigenetic regulation of SMO, bisulfite sequencing PCR (BSP) and methylation-specific PCR (MSP) were carried out to determine the methylation status of the potential SMO promoter region. The relationship between the methylation status and SMO mRNA expression was analyzed.

## 2. Results

### 2.1. Expression of HH Signaling Molecules in Cancer Cell Lines and Normal Prostate Cells and Tissues

To gain a greater understanding of the transcriptional regulation of the HH pathway components, a careful analysis of mRNA levels for key HH signaling genes was conducted. Quantification of mRNA expression levels using a real-time PCR method is increasingly used to determine the activity of HH signaling genes. However, most studies use a relative RT-PCR method, which is less precise and does not provide meaningful comparisons of gene expression between different cell lines. Therefore, accurate quantitation of HH pathway mRNA expression is necessary. In this study, we developed and validated a standard curve based on a Taqman qRT-PCR method to measure key HH signaling genes, including SMO, PTCH1, GLI1, GLI2, and GLI3.

The results produced a broad linear dynamic range of detection of at least six logs and a small quantitative variation produced by triplicate analysis. The slope of the curve was used to determine the reaction efficiency. Efficiency = [10 ^(−1/slope)^] −1 [29]. The efficiency of standard curve for all genes is greater than 92%, and *R^2^* is greater than 0.99.

The expression levels of HH signaling components were determined in 33 cancer cell lines using the quantitative Taqman RT-PCR method. Expression levels of the HH pathway genes, the HH signaling receptors PTCH and SMO, and the target transcription factors GLI2 and GLI3, varied significantly among the cancer cell lines (Figure 1). The SMO gene exhibited the highest level of mRNA expression and the greatest variation between cell lines (22.98 ± 31.80, 95% CI 11.70~31.25), compared with PTCH1 (8.72 ± 10.44, 95% CI 5.02~12.42), GLI2 (1.25 ± 2.08, 95% CI 0.56~1.99) and GLI3 (18.04 ± 20.26, 95% CI 10.85~25.22), whereas GLI1 (0.17 ± 0.25, 95% CI 0.08~0.26) that functions as an amplifier of HH signal, consistently showed low expression in all cancer cell lines. In addition, the absence of SMO expression was confirmed in seven cell lines, including five breast cancer cell lines and the stomach cancer AGS cell line, whereas the lack of SMO was accompanied by undetectable GLI3 in colon cancer HT29 cells. A Significantly lower expression level in normal prostate tissues was confirmed (Table 1).

Significant positive correlations were identified between SMO and GLI2 transcript levels (Pearson’s correlation = 0.359, *p* = 0.040), and between PTCH1 and GLI3 (Pearson’s correlation = 0.532, *p* = 0.001). No correlation was found between the expression levels of SMO/PTCH or SMO/GLI1 (Appendix A).

### 2.2. Hypermethylation of the SMO Gene in Non-Expressing Cancer Cell Lines

We have identified three CpG islands in the 5′-flanking region of the *SMO* gene, and CpG island 1 is located in the proximal promoter region. We therefore analyzed the methylation status of CpG island 1 of the SMO gene in 33 cancer cell lines using the MSP and BSP methods (Figure 2 and Table 2). The sequences and locations of the primer pairs used in BSP and MSP are shown in Appendix A and Figure 2A. The methylation frequency was determined using MSP real time PCR. The seven cell lines that did not express *SMO* (SUM52, MCF10A, MB231, SUN159, MCF7, AGS, and HT29) were all hypermethylated in the amplification region (Table 1 and Figure 2B). In contrast, less than 10% methylation was found in the other cell lines, which expressed *SMO* (Table 2). To determine the methylation status in all three CpG islands, we performed BSP for AGS, MCF7, SKBR3 and PC3 cells. Ten clones of the amplified region of the putative *SMO* promoter for each cancer cell line were sequenced, and methylation status was established for three CpG islands in this region using bisulfite sequencing. The full methylation of all three CpG islands was confirmed in AGS and MCF7 cells, whereas no methylation was found in the cell lines SKBR3 and PC3 that express *SMO* (Figure 2C).

To confirm the role of methylation in silencing *SMO* gene expression, the breast cancer cell line MCF7 that lacked SMO gene expression and the prostate cancer cell line PC3 with moderate *SMO* expression were treated with 5-aza-dC for 72 h Treatment with 5-aza-dC resulted in the expression of *SMO* in MCF7 cells, however, treatment with 5-aza-dC decreased *SMO* expression in PC3 cells (Table 3).

### 2.3. Interspecies Comparison of Genomic SMO Sequences

For a comparison of mammalian *SMO* genes, we analyzed 20,500 bp of *SMO* genome sequence including the upstream; 5′-UTR; exon 1; and part of intron 1 regions for human, mouse, and rhesus monkey. Multiple sequence alignment of the 20,500 bp of *SMO* was performed by the mVista web-tool (Figure 3). The macaque sequence is highly homologous to the human sequence, as 87.5% of the 20-kb region showed at least 88% sequence identity over a 100 bp window. In contrast, the mouse sequence shares 71.2% identity with human, with several conserved noncoding sequences (CNS) showing at least 70% identity over 100 bps.

The global genomic sequence comparison showed significant highly conserved regions among the three genes immediately upstream of the transcriptional start site (*p* = 3.9 × 10^−25^). Five additional CNS were identified, but all had a lower *p* value that was greater than 0.005. A 924 bp region consisting of the full length 5′-UTR and exon 1 regions, along with 293 bp of upstream and 20 bp of intron 1 sequence, is shown in Figure 3B. This region possesses 83% identity between the mouse and human *SMO* genes, and 95% between human and rhesus.

### 2.4. In Silico Analysis of the SMO Upstream Regulatory Region

Submission of a 1611 bp sequence, including 1000 bp 5′-upstream region and the full Exon 1 of the human SMO gene, to the MatInspector software program (core similarity > 0.85; matrix similarity optimized) returned 336 potential TFBS, distributed over the entire sequence. Furthermore, we performed an analysis with the PromoterInspector program and found a 1028 bp potential promoter region within this sequence, located from position −508 to +520 bp. By using the ModelInspector program, 25 models were identified, including SMAD-MIT, SMAD-AP1, YY1-SMAD, ETF-AP1, SP1-ETS. SP1F-NF1, IKRS-AP2, EGR-SP1, SP1-KLFS, GATA-SP1, CAAT-CAAT, NFKB-SP1, and SP1-CAAT.

### 2.5. Functional Analysis of the Core SMO Promoter

Sequence analysis revealed that the 5′-flanking region of the human SMO gene exhibits a high GC content and lacks a consensus TATA element. Three potential CpG islands were identified surrounding the *SMO* gene TSS using the MethPrimer program (Figure 4A). We evaluated different experimental PCR conditions and programs for amplification of the *SMO* promoter sequence. The GC-rich sequences contained within the *SMO* gene promoter region were effectively amplified by a touchdown program in the presence of 3% DMSO [30,31]. Comparison of the touchdown PCR results with PCR under standard condition (fixed annealing temperature, 35 cycles) program for five primer-pairs corresponding to the upstream region of *SMO* is shown in Appendix A and Appendix A.

The potential promoter region upstream of the *SMO* gene was analyzed using interspecies comparison with the Genomatix package. Multiple regulatory elements are located surrounding the TSS of SMO and may play a role in the regulation of *SMO* expression. To determine the minimal sequences required for promoter function and identify cis-acting elements controlling *SMO* promoter activity, a series of truncated luciferase constructs were generated by progressive deletion from the 5′ end of a 984 bp fragment (region from −959 to +25 relative to the TSS), to produce five constructs (Figure 4B and Appendix A), based on our 5′ truncation analysis result (Appendix A). Plasmids containing *SMO* gene fragments were transiently transfected into three cancer cell lines (prostate cancer line PC3; breast cancer lines BT549 and MCF7), and the luciferase activities of these constructs were measured.

The highest promoter activity was observed in MCF7 cells, and moderate activity was found in PC3 and BT549 cells. In MCF7 cells, increased promoter activity was detected upon removal of 459 bp of 5′ sequence up to position −500 bp (relative to the transcription initiation site), indicating the presence of negative regulatory element(s) in the region from −959 bp to −500 bp in MCF7 cells. When truncated to −470 bp, the promoter activity returned to the full-length promoter activity, and deletion of additional sequence to either −400 or −293 further reduced promoter activity, suggesting the presence of positive regulatory element (s) in the region −500 bp to −293 bp. In PC3 and BT549 cells, the 5′ truncations had little effect and maximal activity was observed with the PGL3-SMO-400/+25 construct. The PGL3-SMO-500/+25 construct exhibited the highest promoter activity in MCF7 cells, therefore, this reporter vector was used for subsequent 3′ deletion analysis (Figure 4B).

To further identify the 3′ boundary of the core promoter, three plasmids were generated sharing the same 5′ boundary at position −500, and variable 3′ ends from +50 to −15. In contrast to the results from the 5′ deletion analysis, luciferase activity with the 3′ deletions showed similar effects in all three cell lines. The promoter activities were comparable between the PGL3-SMO-500/+25 and PGL3-SMO-500/+15 in the three cell lines, while the addition of 25 bp of 3′ sequence to +50 resulted in a decreased activity (Figure 4B), indicating an absence of downstream promoter element activity in the *SMO* promoter.

Taken together, these results demonstrated that the region between −500 and +15 bp is important for the transcriptional activity of the *SMO* promoter, and both negative and positive regulatory regions can affect the promoter activity of the *SMO* gene depending on cell context.

### 2.6. Identification of Transcription Factor Binding Sites in the SMO Gene Promoter

EMSA experiments were conducted to investigate the binding of nuclear proteins to the core proximal *SMO* promoter sequences in nuclear extracts of PC3, BT549 and MCF7 cells. (Figure 5). Six overlapping oligonucleotide probes covering the region between −500 and −357 bp (Figure 5A) that significantly enhanced promoter activity in MCF7 cells were prepared to investigate their DNA-protein binding activity. Double-stranded DNA probes spanning ~30 bp, covering the regions −500 to −471 (SMO-500P); −472 to −444 (SMO-472P); −454 to −421 (SMO-454P); −422 to −398 (SMO-422P); −400 to −373 (SMO-400P); and −383 to −357 (SMO-383P) were prepared. The SMO-472P, SMO-454P, SMO-422P, and SMO-383P probes showed clear binding with nuclear proteins extracted from PC3, BT549 and MCF7 cells, while the SMO-500P and SMO-400P probes did not produce strong complexes with nuclear proteins from any of the cell lines tested (Figure 5B). The SMO-472P, SMO-422P and SMO-383P showed strong binding with nuclear protein from MCF7, but strong binding in the SMO-454P region was observed with nuclear proteins from PC3 and BT549 (Figure 5B).

The specificity of binding was tested in competition experiments using excess unlabeled oligonucleotides carrying consensus sequences and specific antibodies. A competition assay was conducted with four probes, SMO-472P, SMO-454P, SMO-422P, and SMO-383P, to confirm DNA-protein complexes. The DNA-protein complexes formed by the SMO-472P and SMO-422P probes were reduced or disappeared completely in the presence of a 50-fold excess of the consensus SP1-binding oligonucleotide, but not in the presence of excess unlabeled consensus oligonucleotides for other TFs (Figure 5C). The major complex formed by the SMO-454P probe disappeared completely in the presence of a 50-fold excess of unlabeled consensus AP1 and was greatly reduced by CREB oligonucleotides, but not by the addition of excess unlabeled consensus SP1 and AP-2α oligonucleotides. The major band observed with the SMO-383P probe was reduced only in the presence of a 50-fold excess of unlabeled consensus AP-2α oligonucleotide (Figure 5C).

A supershift assay with specific antibodies was performed using nuclear extracts from PC3 and MCF7 cells to confirm the identity of the TFs generating the complexes observed in Figure 5C. As shown in Figure 5D, the DNA-protein complexes in the SMO-472P and SMO-422P regions were supershifted by the anti-SP1 antibody. The complex in the SMO-383P region was supershifted by the anti-AP-2α antibody. The complex in the SMO-454P region was inhibited by anti-c-Jun, anti-ATF1, and anti-CREB antibodies. Therefore, SP1, AP1 (c-Jun/ATF1)/CREB, and AP-2α all appear to play a role in *SMO* gene regulation.

## 3. Discussion

The HH pathway drives oncogenesis in many cancers, and strategies targeting this pathway have been developed, most notably through inhibition of SMO, which is a key step involved in the regulation of the seven-transmembrane oncoprotein. SMO can activate the glioma-associated oncogene (GLI) family of transcription factors, leading to hyperproliferation of epithelial cells [32].

CpG islands represent a common epigenetic element that regulates transcription at many promoters through methylation-induced silencing. In this study, the most striking feature revealed by in silico analysis of the *SMO* promoter is the abundance of CpG dinucleotides and multiple SP1 binding sites (7 × SP1 sites) close to the TSS. SP1 has been widely described as a general transcription factor involved in the transcription of gene promoters that lack a TATA box. CpG-rich promoters bound by DNA sequence-specific transcription factors including SP1 have the highest expression level, and deletion of SP1 binding sites results in significantly decreased promoter activity [33,34].

We have isolated and cloned DNA fragments containing the predicted *SMO* promoter region, and the strongest promoter activity was identified in the 5′-UTR region around −500/+25 that contains a high GC content, is CpG rich and lacks a canonical TATA box. Evaluation of the predicted SP1 consensus sites revealed that the SMO-472P (−472 to −444 bp) and SMO-422P (−422 to −398 bp) regions are functional in binding to the SP1 transcriptional factor as shown in EMSA/supershift experiments. In addition, inducible TF factors binding to AP1 and AP2 sites were also identified in the promoter region and may play a role in modulating SMO expression.

The epigenetic regulation of *SMO* transcription was characterized in 33 cancer cell lines. We determined the mRNA expression of the major HH pathway genes, *SMO*, *PTCH1*, *GLI1*, *GLI2*, and *GLI3* and found the highest level of mRNA expression was observed in the *SMO* gene, but their corresponding proteins were very rare or produced weak signals in normal breast epithelium [35]. We have also measured mRNA expression levels of *SMO*, *PTCH1* and *GLI1* in the NCI60 cell line panel (Appendix A). *SMO* expression was undetectable in 8 of 60 cancer cell lines in the panel. The highest expression of *SMO* was found in ovary, followed by lung and kidney cancer cell lines. Fagerberg, L. et al., have carried out a comprehensive analysis by RNA-seq and combined antibody-based proteomics to classify the tissue-specific expression of genes across major human organs and tissues [36]. Expression of SMO is variable and can be detected in all 27 tissue samples. The highest expression of SMO was observed in ovary, endometrium, skin, and prostate. In addition, expression in normal human tissues obtained through the UCSC genome browser exhibited similar patterns: expression of SMO and GLI3 is high overall, but variable, whereas PTCH1 and GLI1 seem to have a very selective expression pattern. Analysis of the expression levels of the HH pathway genes in the 1457 cell lines in the Cancer Cell Line Encyclopedia (CCLE, https://portals.broadinstitute.org/ccle) showed a significant positive correlation (r = 0.37, *p* < 0.0001) between *SMO* and *GLI2*, supporting the conclusion that SMO gene regulation is important in HH pathway regulation (Appendix A).

DNA methylation is a major epigenetic regulatory mechanism of gene expression and is involved in the progression of cancer [37]. The absence of *SMO* expression in seven cell lines derived from breast (5 of 9), stomach (1 of 1), and colon (1 of 1) cancer tissues was correlated with a high level of gene methylation. The level of *SMO* mRNA was negatively correlated with the methylation status of the *SMO* promoter (Figure 6). In this study, *SMO* and *GLI3* were undetectable in the HT29 colon cancer cell line. This result is consistent with a previous study that showed SMO methylation leads to the silencing of *GLI3* expression [38]. Moreover, full methylation was confirmed in all three CpG islands in the MCF7 breast cancer cell line, and stomach cancer cell line AGS, and *SMO* expression in MCF7 cells were restored after 1 µM 5-Aza treatment. The results indicate that DNA methylation of the SMO gene may play an important role in the development of cancer. Cell lines from breast cancer tissue showed the highest methylation frequency, 56% (5/9), whereas eight prostate cancer cell lines had no detectable methylation in the *SMO* gene. Whether the degree of *SMO* methylation correlates with the tissue specificity remains to be explored. By using a ChIP-sequencing approach, specific histone mark Histone 3 Lysine 4 Acetylation (H3K4Ac) peaks have been confirmed in the proximal promoter of the *SMO* and *GLI1* genes, demonstrating the expression of these genes was regulated by the removal of H3K4Ac mediated by Histone Deacetylase 3 (HDAC3) [39]. It will be important to perform functional assays to validate transcription factors binding to *SMO* promoter region and their biological impact using ChIP-seq or site-directed mutagenesis in future studies.

The full mechanistic details of HH signal transduction are still under investigation. Abnormal HH activation has been implicated in tumorigenesis in a wide variety of tumors, and SMO and GLI play a critical role in this pathway. GLI2 is suggested to function primarily as a transcriptional activator, and GLI3 as a repressor [4]. A significant positive correlation was identified between the expression of *SMO* and *GLI2* in our study. This is consistent with previous reports that identified overexpression of SMO and GLI2 in progesterone receptor (PR) negative breast cancers and gastric cancers [35,40]. Overexpression of GLI1 and GLI2 leads to tumor development in transgenic mice, suggesting that GLI1 or GLI2 contribute to tumorigenesis [17,41]. *PTCH1* is an established tumor suppressor gene and developmental regulator. Although the role of GLI3 as a negative regulator of HH signaling is well established in the context of normal development, its role in cancer has largely been ignored [42,43,44]. In this study, we have demonstrated that *GLI3* expression is positively correlated with *PTCH1* levels. This result is indirectly supported by the finding of increased expression of PTCH and GLI3 in cancerous tissues and correlated with the increased proliferating index of Ki-67 in breast cancer. Moreover, *PTCH1* haploinsufficiency is associated with distinct autosomal dominant syndromes [45]. Previous studies have demonstrated that loss of function mutations of the tumor suppressor PTCH1 or gain of function mutations of SMO are associated with basal cell carcinoma [46,47]. These data suggest that the repressive effect of PTCH1 and GLI3 on HH signaling has a crucial role in cancer development.

## 4. Materials and Methods

### 4.1. Computational Analysis of the SMO Gene

Identification and sequence analysis of evolutionarily conserved regions (ECRs) of the *SMO* gene were performed with the ECR Browser, and the publicly available web-based tool mVista [48] using the MLAGAN algorithm. A search for potential TFBS in the upstream regulatory region of the SMO gene was performed online at Genomatix using the MatInspector program (http://www.cbrc.jb/research/db/TFSEARCH.html).

### 4.2. Cell Lines and 5-Aza Treatment

All cell lines were purchased from American Type Culture Collection (ATCC, Rockville, MD, USA) and grown according to the ATCC instructions. Total RNA from human prostate tissues was obtained from Clontech (Mountain View, CA, USA). 5-Azacytidine (5-Aza, Sigma-Aldrich, St. Louis, MO, USA) was freshly prepared in PBS before use. A vehicle control consisting of culture medium alone was included in the analysis. MCF7 and PC3 cells were pre-cultured for 24 h, then treated with 1 µM 5-Aza for 72 h. Cells were collected by centrifugation, then genomic DNA and RNA were extracted and analyzed.

### 4.3. Real Time Quantitative RT-PCR

Total cellular RNA was isolated and purified by RNeasy columns (QIAGEN Valencia, CA, USA) according to the manufacturer’s instructions with on-column and in-solution DNaseI digestion. RNA quality and quantity were determined using Agilent RNA 6000 Nano Chip (Agilent Technologies Inc., CA, USA). cDNA synthesis was carried out using Random Hexamer primer, Taqman Reverse Transcription Reagents kit (Applied Biosystems Foster City, CA, USA). Taqman real time RT-PCR primers and probes for target genes were designed by using the Primer Express software. SMO Fwd: 5′-GAGACTCGGACTCCCAG-3′; Rev: 5′-GTATACGGCACACAGCAG-3′ and probe: 5′(FAM)-TCGGGCCTCCGGAAT-(MGB)3′. PTCH1 Fwd: 5′-GCATAGGAGTGGAGTTCA-3′; 5′-CCCTGCGGTTCTTGTC-3′ and probe 5′(FAM)-TTGGCCTTTCT-(MGB)3′. GLI1 Fwd: 5′-GTCTCAAACTGCCCAGC-3′; Rev: 5′-CGTTCAAGAGAGACTGGG-3′ and Probe: 5′(FAM)-TCCCACACCGGTACCA-(MGB)3′. PTCH2, (Assay ID = Hs01085642_ml), GLI2 (Assay ID = Hs00257977_ml) and GLI3 (Assay ID = Hs00609233_ml). Taqman real time RT-PCR was used to determine the expression profile, with a 18S rRNA plasmid as the standard reference gene using primers Fwd: 5′-CCGAAGCGTTTACTTTGAAAAAA-3′; Rev: 5′-TTCCATTATTCCTAGCTGCGGTAT-3′ and probe 5′(VIC)-AGTGTTCAAAGCAGGCC-(MGB)3′. The PCR reactions were performed in 20 µL final volume containing 5 ng of cDNA, 1 × Master Mix (TaqMan Universal PCR Master Mix, ABI, CA, USA), 900 nM of each primer and 200 nM of each probe, respectively. The thermal cycling conditions are 40 cycles of PCR amplification (UNG incubation: 50 °C, 2 min; Ampli TaqGold activation: 95 °C, 10 min; denaturation: 95 °C, 15 s; annealing/extension: 60 °C, 1 min) (ABI PRISM 7900HT Sequence Detection System, CA, USA). All assays were performed in triplicate, and each plate contained the same standard and positive quality control sample. For each unknown sample, the copy number of each gene is calculated using linear regression analysis from their respective standard curves. The relative mRNA expression level of target genes was normalized by the following formula: (copy number of target gene)/(copy number of 18S rRNA) × 10e6. The standard curves were generated using a dilution series of plasmids containing *SMO*, *PTCH1*, *PTCH2*, *GLI1*, *GLI2*, and *GLI3* from full length cDNA (ATCC, Rockville, MD, USA). The copy number of plasmid cDNA was calculated by optical density according to the exact molar mass derived from the sequences. Serial dilutions were made to obtain 10e1 to 10e7 copies. The slope and intercept were calculated for each run using a linear regression analysis of the log copy number versus threshold cycle (Ct) value for both target genes and 18S rRNA standard curves [49].

### 4.4. Bisulfite Modification and Bisulfite Sequencing PCR (BSP)

DNA was extracted using the QIAamp DNA Mini Kit (Qiagen, Valencia, CA, USA). Bisulfite modification of 1µg of genomic DNA was performed with the EpiTect Bisulfite Kit (Qiagen, Hilden, Germany) as described by the manufacturer. Primers for BSP and identification of predicted CpG islands in the SMO promoter region were carried out with the assistance of Methyl Primer Express Software v1.0 (Applied Biosystems) and MethPrimer (http://www.urogene.org/methprimer/). The primer sequences used for methylation analysis are summarized in Appendix A. PCR reactions were performed in a volume of 25 µL containing 10 ng of bisulfite-converted DNA and 20 pmol of each primer using Platinum^®^ PCR SuperMix (Invitrogen, Carlsbad, CA, USA). Thermal cycling conditions were 95 °C for 2 min, followed by 35 cycles of 95 °C for 15 s, 56 °C for 30 s, 72 °C for 10 s, and a final extension at 72 °C for 5 min. For sequence analysis, the PCR products were subcloned into a pCR2.1 vector using a TOPO TA Cloning kit (Invitrogen, Carlsbad, CA, USA) according to the manufacturer’s instructions. At least 12 clones were sequenced in an ABI 3730 DNA Sequencer (Applied Biosystems, Foster City, CA, USA) for each cell line tested. Methylation analysis was performed using BiQ Analyzer software [50].

### 4.5. Methylation-Specific PCR (MSP)

The MSP products were 132 bp long. Unmethylated and methylated human DNA were used as a negative and a positive control, respectively (Qiagen, Hilden, Germany). Water blanks and PCR mixtures without template were also used as experimental controls in each assay. The primers were listed in Appendix A. The amplification cycles performed were 38 cycles. After PCR, products were separated on a 2% agarose gel, and stained with ethidium bromide. Bisulfite treatment and MS-PCR assays were performed in duplicate for all samples. Each experiment was performed at least three times.

### 4.6. Real-Time Quantitative MSP

The bisulfite–converted genomic DNA was amplified using fluorescence-based real-time MSP using FastStart SYBR Green Master Kit (Roche). Methylation of the *SMO* gene was examined using actin as the internal control for DNA quantification. The beta actin gene contains no CpG dinucleotides and is not affected by DNA methylation status or bisulfite treatment. The primers for quantitative MSP are the same as the normal MSP shown in Table 1. Real-time PCR conditions were 95 °C for 10 min followed by 40 cycles of 94 °C for 15 s, 59 °C for 60 s with data acquisition after each cycle. In the end, properties of real-time PCR conditions and amplification products were checked by melting curve analysis. PCRs were done in two replicates of each sample with the 7900HT Fast Real-Time PCR System (Applied Biosystems).

### 4.7. Touchdown PCR of the SMO Promoter Region

PCR was carried out in a volume of 50 µL containing 100 ng of genomic DNA, 20 pmol of each primer and 3% Dimethyl Sulfoxide (DMSO, Sigma-Aldrich, St. Louis, MO, USA) using Platinum^®^ PCR SuperMix Kit (Invitrogen, Carlsbad, CA, USA). A modified touchdown PCR was performed with the following cycling conditions: The templates were denatured at 94 °C for 3 min, and then 20 cycles composed of 20 s at 95 °C, 30 s annealing with a stepwise reduction of annealing temperature from 68 °C to 58 °C decreasing by 0.5 °C every cycle, and an elongation step of 4 min at 72 °C. Twenty additional cycles were then performed at 94 °C 20 s, 58 °C for 40 s, and 72 °C for 40 s. The standard PCR program was 35 cycles at 94 °C 20 s, 58 °C for 40 s, and 72 °C for 40 s. All PCR products were analyzed by electrophoresis on a 1.0% agarose gel stained with ethidium bromide.

### 4.8. Generation of Luciferase Reporter Plasmids

A series of truncated *SMO* promoter constructs, including five deletions from the 5′ side and three deletions on the 3′ side, were created by PCR using the primers shown in Appendix A. PCR products were cloned into the TOPO-TA vector, and inserts were excised with SacI and XhoI and cloned into pGL3 (Promega, Madison, WI, USA) to generate constructs in the forward orientation. All subclones were verified by sequencing. Sequence analysis was performed with the Molecular Evolutionary Genetics Analysis (MEGA) software version 7.

### 4.9. Cell Transfection and Luciferase Assays

Two breast cancer cell lines, MCF7 and BT549, and prostate cancer cell line PC3, were used for the analysis of promoter constructs. The cells were plated at 1 × 10^5^ cells per well in a 24-well plate the day before transfection and incubated overnight at 37 °C in 5% CO_2_. For each well, 5 µL of HilyMax transfection reagent (Dojindo, Rockville, MD, USA) was diluted in 30 µL of growth medium without serum and incubated at room temperature for 5 min. The DNA mixture containing 2 µg of the specific reporter construct plus 20 ng of Renilla luciferase pRL-SV40 control DNA was then added to each well, and incubated at room temperature for 20 min. Luciferase activity was assayed at 48 h using the Dual-Luciferase Reporter Assay System (Promega, Madison, WI, USA) according to the manufacturer’s instructions. Measurement of the firefly luciferase activity of the SMO promoter constructs was normalized relative to the activity of the Renilla luciferase produced by the pRLSV40 control vector and each construct was tested in triplicate in at least three independent experiments.

### 4.10. Electrophoretic Mobility Shift Assays (EMSA)

Nuclear extracts were prepared from PC3, BT549 and MCF7 cells using the CellLytic NuCLEAR extraction kit (Sigma-Aldrich, St. Louis, MO, USA). Six double-stranded DNA oligonucleotide probes corresponding to −500 to −357 bp of SMO promoter were synthesized (Figure 5A, upper panel). Labeling, DNA-protein binding reactions and antibody supershift experiments were performed as previously described [49,51]. For supershift experiments, antibody was added after the addition of labeled DNA-probe, and the binding reaction was incubated for an additional 20 min at room temperature. For competition analyses, a 50-fold excess of consensus unlabeled-competitor oligonucleotides, SP1, AP1, Ap-2α, and CREB (Santa Cruz Biotechnology, Santa Cruz, CA, USA), were included in the binding reaction.

### 4.11. Statistical Analysis

The correlation of mRNA expression levels of *SMO*, *PTCH1*, *GLI1*, *GLI2*, and *GLI3* were assessed by Pearson’s correlation coefficient using GraphPad Prism 7 software. All *p* values reported were two-tailed, with significance defined as *p* < 0.05.

## 5. Conclusions

In conclusion, our data provide strong experimental and computational evidence for genetic and epigenetic regulatory mechanisms of the *SMO* gene. The SMO promoter has been characterized and its major regulatory elements, including multiple CpG islands and SP1 binding sites, were identified. A correlation between *SMO*/*GLI2* and *PTCH1*/*GLI3* expression was observed. Moreover, SMO expression is correlated with the degree of CpG island methylation. Our results reveal a central role for epigenetic regulation of *SMO* gene transcription that may be exploited for the development of new therapeutic strategies to treat hedgehog-driven tumors.

## Figures and Tables

**Figure 1 cancers-12-02219-f001:**
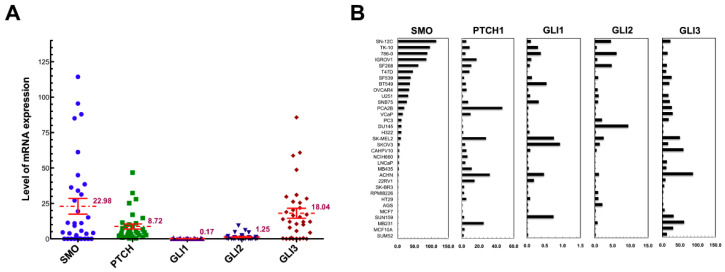
The expression levels of HH signaling components. (**A**) Distribution of mRNA levels of SMO, PTCH, GLI1, GLI2, and GLI3 in 33 cancer cell lines. Mean values ± SCE of each gene are indicated by horizontal bars. (**B**) Comparison of mRNA levels. The mRNA levels of the five genes were quantitated from the Taqman RT-PCR as described in Materials and Methods. Data are presented by cell line in decreasing order of SMO mRNA level.

**Figure 2 cancers-12-02219-f002:**
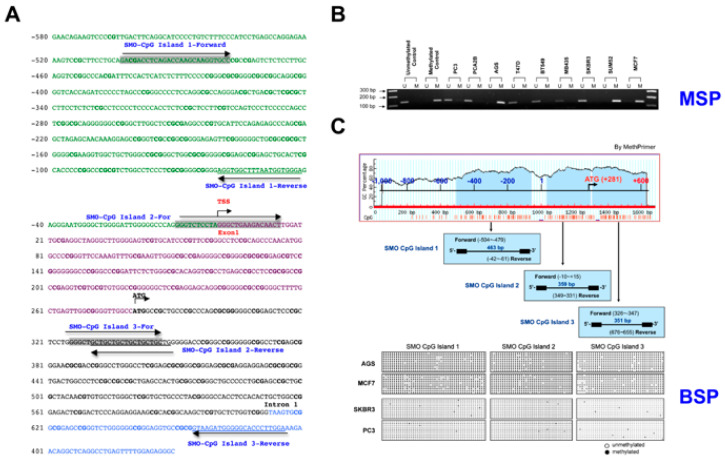
Hypomethylation of the 5′-flanking region of the *SMO* gene. (**A**) Bases are numbered relative to the transcription start site at position + 1. CpG sites are shown in bold. The primers used for amplification and sequencing of bisulfate modified DNA were indicated by grey for forward and underline for reverse primer. The long arrows indicate the orientation. (**B**) Methylation-specific PCR analysis of the *SMO* upstream regulatory region in methylated/unmethylated controls and nine cancer cell lines. M indicates hypermethylated SMO; U indicates unmethylated *SMO*. (**C**) *SMO* promoter methylation analysis by MethPrimer. Three CpG-rich regions surrounding SMO TSS in a span of the 1611 base pairs and results of bisulfite DNA sequencing were shown.

**Figure 3 cancers-12-02219-f003:**
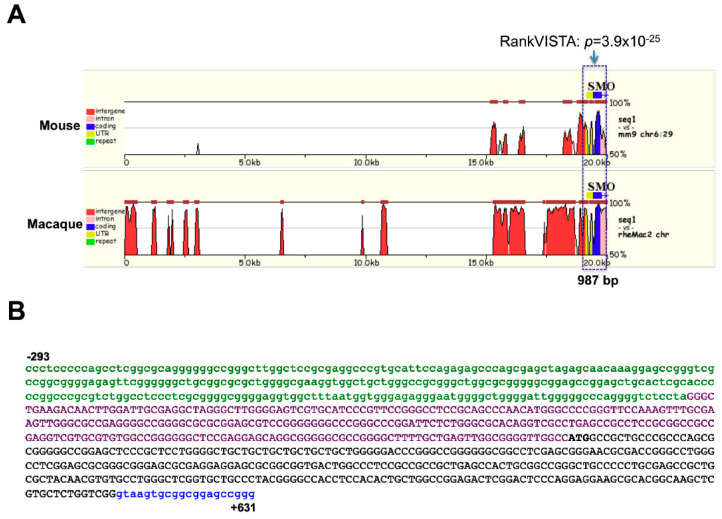
In silico analysis of the *SMO* upstream region. (**A**) Alignment of the 5′-flanking region of three mammalian *SMO* genes. A 20 kb segment of upstream sequence including exon-1 from the mouse, macaque and human *SMO* genes was aligned by the MLAGAN algorithm of the mVista program. The sequences of mouse and macaque are aligned to the human SMO sequence (*x*-axis); numbering is relative to the transcription start site. Conserved regions (>70% homology over 100 bp window) are shaded. The box indicates conserved regions among the three sequences, as determined by RankVista (*p* ≤ 10^−5^), with the *p* values given above. (**B**) The structure of the 924 bp conserved sequence. The 5′-flanking region is defined as sequence upstream (from the 5′ end) of the transcript start site and shown in green lower-case letters. The intron is shown in blue lower-case letters. The exon is shown in uppercase letters, with UTR as purple and coding sequence as black color, respectively.

**Figure 4 cancers-12-02219-f004:**
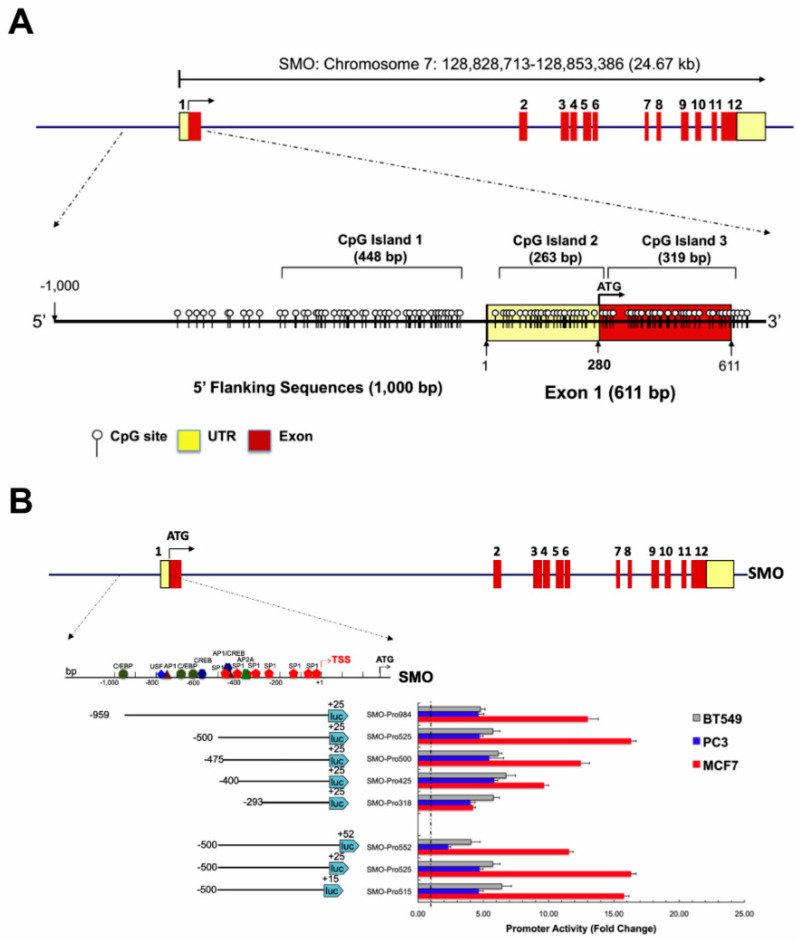
Functional analysis of the core *SMO* promoter. (**A**) Structure of the *SMO* gene. A schematic representation of the exon-intron organization and UTR region. Twelve exons are indicated by the numbered rectangles. Distribution of CpG dinucleotides in a 1611 bp fragment of the *SMO* gene harboring 1000 bp 5′ upstream region and full exon 1 is shown. Each vertical line represents a single CpG site. Numbering is relative to the transcription start site at exon 1. Transcription orientations are indicated by arrows. (**B**) Functional localization of the *SMO* promoter. A schematic of the *SMO* gene structure is shown above. Twelve SMO exons are indicated by the numbered rectangles. A schematic diagram of the 1500 bp 5′-flanking region of *SMO* and serial truncation constructs of the SMO promoter and their corresponding luciferase activities in different cell types are shown. Serial deletions at the 5′ and the 3′ ends of the promoter fragment of *SMO* are shown on the left. The promoter activities measured after transfection into PC3, BT549 and MCF7 cells are shown on the right. The relative size and position of fragments cloned into the pGL3 vector are indicated by the lines below the schematic, and the numbers in parentheses on either side of each fragment indicate the distance in nucleotides upstream from the *SMO* start codon of the 5′ and 3′ ends of each fragment. The luciferase activity of the pGL3 constructs is shown as fold-increase of corrected light units relative to an empty pGL3 vector control. Values represent the mean, and error bars indicate the SEM of at least three independent experiments.

**Figure 5 cancers-12-02219-f005:**
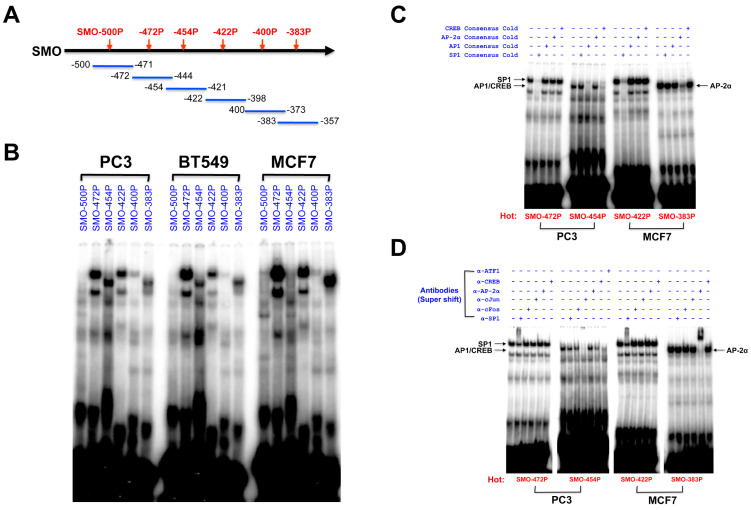
EMSA analysis of the core region of SMO promoter. (**A**) A binding assay using overlapping probes in the *SMO* promoter. A schematic illustration of six probes used for EMSA is shown in the upper panel. (**B**) The 32P-labeled probes were incubated with nuclear extracts from PC3, BT549 and MCF7 cells, respectively. (**C**) Competition analysis using a 50-fold excess of unlabeled oligonucleotides (cold). The 32P-labeled SMO-472P and SMO-422P were incubated with nuclear extracts from PC3 cells in the presence of a 50-fold excess unlabeled consensus SP1, AP1, AP-2α, and CREB oligonucleotides, respectively. Similarly, the competition analysis of 32P-labeled SMO-454P and SMO-383P were performed with nuclear extracts from MCF7 cells. (**D**) Supershift analysis using specific antibodies. Antibodies (2 µg), including anti-SP1, anti-c-Fos, anti-c-Jun, anti-AP-2α, anti-CREB and anti-ATF1, were preincubated with 10 µg of nuclear extracts before the addition of the 32P-labeled probes. The bands of SP1, AP1/CREB, and AP-2α binding are indicated by arrows.

**Figure 6 cancers-12-02219-f006:**
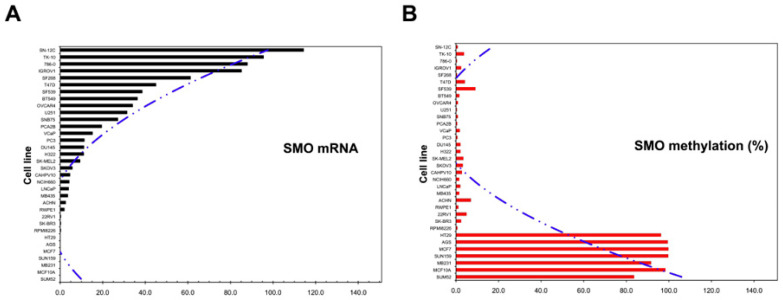
Correlation of mRNA level and methylation frequency of the *SMO* gene in 33 cancer cell lines. (**A**) mRNA level of *SMO* in decreasing order. (**B**) Distribution of methylation frequency (%) in the cell lines by a quantitative MSP method.

**Table 1 cancers-12-02219-t001:** The expression level of hedgehog pathway genes in 33 tumor cell lines and 5 normal cells and tissues.

Level of mRNA Expression *											
		SMO		PTCH		GLI1		GLI2		GLI3	
Tissue	Cell lines	Mean	SD	Mean	SD	Mean	SD	Mean	SD	Mean	SD
Prostate	PCA2B	19.46	1.34	46.74	4.84	0.03	0.01	0.00	0.00	26.22	7.56
	VCaP	15.15	1.13	10.15	0.99	0.00	0.00	0.10	0.03	28.43	8.59
	PC3	11.33	0.59	0.39	0.04	0.02	0.01	1.96	0.13	17.13	2.73
	DU145	11.14	1.88	1.27	0.15	0.03	0.01	9.36	0.58	0.47	0.09
	CAHPV10	4.59	0.37	5.31	0.49	0.08	0.02	0.43	0.08	58.71	5.91
	NCIH660	4.18	1.00	6.61	1.41	0.01	0.00	ND		0.02	0.01
	LNCaP	3.90	0.45	4.10	0.51	0.01	0.01	ND		12.09	0.85
	22RV1	0.18	0.10	14.68	0.13	0.19	0.04	0.01	0.01	7.91	0.76
Breast	T47D	44.96	1.23	8.71	0.53	ND		ND		9.99	0.76
	BT549	36.26	5.84	4.64	0.69	0.54	0.10	ND		18.91	3.93
	MB435	3.56	0.50	11.43	1.99	0.04	0.02	ND		10.56	1.76
	SK-BR3	0.02	0.00	2.72	0.32	0.00	0.01	0.00	0.00	4.39	1.21
	SUM52	ND		2.16	0.85	0.02	0.00	0.00		12.09	3.09
	MCF10A	ND		3.24	0.68	ND		ND		29.75	0.55
	MB231	ND		25.20	1.99	ND		0.71	0.07	60.85	2.76
	SUN159	ND		2.68	0.59	0.74	0.13	ND		31.01	0.70
	MCF7	ND		0.87	0.16	0.03	0.01	0.01	0.00	5.54	0.63
Kidney	SN-12C	114.33	8.01	5.11	1.19	0.10	0.04	4.47	0.21	21.85	2.9
	TK-10	95.49	9.04	8.95	3.19	0.30	0.13	0.46	0.04	0.08	0.01
	786-0	87.90	6.39	4.22	1.05	0.38	0.12	6.07	0.73	13.87	3.93
	ACHN	2.54	0.25	32.37	1.43	0.47	0.05	1.09	0.45	85.73	3.36
Glioblastoma	U251	31.36	4.97	1.19	0.19	0.08	0.01	1.01	0.10	16.50	1.16
	SF539	38.50	4.61	2.61	0.77	0.13	0.05	0.85	0.09	25.44	1.65
	SNB75	27.07	4.50	7.33	0.82	0.32	0.07	0.83	0.09	20.28	0.59
	SF268	61.21	9.89	10.97	2.55	0.07	0.02	4.70	0.25	11.96	0.57
Ovary	IGROV1	85.11	13.96	17.00	3.53	0.11	0.01	0.62	0.02	ND	
	OVCAR4	33.99	3.56	5.46	1.15	0.03	0.02	0.74	0.08	ND	
	SKOV3	5.65	1.48	3.85	0.57	0.92	0.07	1.32	0.19	15.48	1.24
Others											
Stomach	AGS	ND		0.37	0.06	0.01	0.00	2.07	0.91	0.08	0.004
Skin	SK-MEL2	9.28	0.85	27.98	6.50	0.75	0.16	2.43	0.10	48.77	5.6
Colon	HT29	ND		5.09	0.30	0.08	0.01	0.93	0.04	ND	
Lung	H322	11.00	2.63	1.64	0.16	0.06	0.03	0.19	0.01	0.75	0.17
Myeloma	RPMI8226	0.02	0.00	2.64	0.46	0.02	0.01	0.85	0.09	0.37	0.07
Normal prostate cell	WPMY	30.23	6.66	4.39	1.00	0.44	0.10	14.73	4.23	51.85	14.58
Normal prostate cell	WPE-sterm	33.51	1.35	5.40	0.45	0.13	0.01	0.47	0.06	88.00	8.36
Normal human prostate tissue		7.71	2.90	2.93	1.21	0.18	0.01	0.44	0.02	9.87	0.41
Normal human trachea tissue		2.89	0.23	1.55	0.06	0.24	0.10	0.65	0.25	6.65	2.83
Normal human breast tissue		2.97	0.56	2.37	0.38	0.23	0.13	0.85	0.35	28.90	10.17

Abbreviations: SD, Standard Deviation; ND, not detectable. * Relative mRNA expression = (Target gene/18s rRNA) × 10^6^.

**Table 2 cancers-12-02219-t002:** Epigenetic regulation of *SMO* mRNA expression in tumor cell lines.

Tissue	Cell Lines	*SMO* mRNA Level *	*SMO* Methylation Frequency (%)
Prostate	PCA2B	19.46	0.47
	VCaP	15.15	1.65
	PC3	11.33	0.61
	DU145	11.14	1.99
	CAHPV10	4.59	2.67
	NCIH660	4.18	1.41
	LNCaP	3.9	1.93
	22RV1	0.18	4.9
Breast	T47D	44.96	4.05
	BT549	36.26	1.47
	MB435	3.56	1.41
	SK-BR3	0.02	2.32
	SUM52	ND	83.57
	MCF10A	ND	98.28
	MB231	ND	91.52
	SUN159	ND	99.6
	MCF7	ND	99.66
Kidney	SN-12C	114.33	0.71
	TK-10	95.49	3.7
	786-0	87.9	0.33
	ACHN	2.54	6.88
Glioblastoma	SF268	61.21	0.36
	SF539	38.5	9.06
	U251	31.36	0.31
	SNB75	27.07	0.77
Ovary	IGROV1	85.11	2.37
	OVCAR4	33.99	0.84
	SKOV3	5.65	3.1
Stomach	AGS	ND	99.36
Skin	SK-MEL2	9.28	3.39
Colon	HT29	ND	96.22
Lung	H322	11	1.98
Myeloma	RPMI8226	0.02	0.57

Abbreviations: ND, not detectable. * Relative mRNA expression = (Target gene/18s rRNA) × 10^6^.

**Table 3 cancers-12-02219-t003:** Demethylation treatment restored *SMO* mRNA expression in MCF7 cell.

	Control		5-Aza Treatment (1 µM)	
Cell Line	Methylation Frequency (%)	mRNA Level	Methylation Frequency (%)	mRNA Level
MCF7	99.6	0	57.9	16.1
PC3	0.6	113.3	1.5	86.5

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
