# Peer review of "Genetic and Epigenetic Regulation of the Smoothened Gene (SMO) in Cancer Cells"

_cancers, 2020, doi:10.3390/cancers12082219_

Round 1
Reviewer 1 Report
In this study, the authors aimed to investigate the epigenetic mechanisms involved in the regulation of expression of the key Hedgehog effector and drug target Smoothened (SMO). Lou et al. analyzed SMO mRNA expression by qPCR in a total of 33 cancer cell lines. In addition to SMO, Lou et al determined the mRNA levels of canonical HH regulators and target genes and performed correlation analysis.
The differences in SMO mRNA expression levels prompted the authors to investigate the cis-regulatory regions and the methylation status of several CpG islands upstream of the SMO gene. Using EMSA assays, they found in vitro binding of several transcriptional regulators including SP1, AP1 and CREB to the putative promoter/enhancer region of SMO. Three hypermethylated CpG islands may account for low expression of SMO in several of the cancer cell lines investigated.
The enthusiasm about the novelty of the findings is very limited. Most of the expression data could have been analyzed in a much more comprehensive manner by studying public data resources such as the Broad cell line data panel comprising RNAseq data of more than 1700 cell lines. The experimental approach to address the regulatory mechanisms involved in SMO expression are also not as appropriate as they could be to draw reasonable conclusions. The data of luciferase reporter assays are also not in line with the expression levels determined by qPCR. For instance, while SMO is basically undetectable in MCF7 cells, luciferase reporter assays show the highest SMO expression level in MCF7 cells. To figure out the regulatory transcription factors controlling SMO expression, which would be of high interest, site-directed mutagenesis and genetic manipulation of the respective regulatory factors would be required to demonstrate the functionality of the binding sites reported in the manuscript. In addition, ChIP analysis would be required to validate binding of the transcription factors to the sites in the cis-regulatory region.
The methylation analysis is technically sound, but what does this actually tell, given the fact that only cell lines have been used in this study? Is this of any clinical relevance? Is the methylation of SMO CpG islands a predictive marker to discriminate responders and non-responders when SMO inhibitors are used? There is some evidence in AML that DAC treatment establishes SMOi sensitivity, however rather by upregulating GLI3 rather than SMO.
The mainly descriptive nature of the manuscript, the lack of sufficient novelty and clinical and mechanistic relevance dampen the reviewer´s support to recommend this study for publication in Cancers.
Author Response
Thank you very much for your comments. Please see our responses in the attachment

Reviewer 2 Report
The manuscript of Lou et al. can be divided into two aspects: 1- expression levels of Hh pathway components in 33 cancer cell lines of different tissues, in which SMO expression is inversely regulated by CpG methylation of its 5' flanking genomic DNA, and 2- analysis of SMO's promoter region and identification of binding sites for SP1, AP1, AP2a and CREB. Both findings are useful and either novel or more widespread (for example, it was known that breast cancer cell lines express very different levels of SMO). However, the study is limited in integration of the findings and makes some surprising conclusions that are not derived form this work nor the literature.
A list of concerns/comments (in order of appearance):
1- It is surprising that all the cancer cell lines selected show low activation of the canonical Hh pathway, as evidenced by very low levels of Gli1 mRNA. This fact suggests that the expression level of SMO, although very variable, does not correlate with signalling outcomes. Could the authors include 2-3 cell lines or primary tumours with LOF of PTCH1 or GOF of SMO, such as some medulloblastoma cell lines or primary basal cell carcinomas? A very high expression of GLI1 in those samples would be re-assuring. They should also investigate the levels of all those mRNAs in a few normal cell lines of the same tissues.
2- In the Methods corresponding to the qRT_PCR, it says that Fig 1S shows the calibration curves with known quantities of cDNA. However, that figure is missing (Fig 1S shows the correlation between expression of different pairs of genes). The Methods also refer to linear analysis of 101 to 107 copies, when it should say 10e1 to 10e7. The manuscript is full of exponentials not superscripted and greek letters showing like @ (presumably mu, for micro).
3- I think the study will be clearer if the expression levels are followed by the methylation analysis. Putting the promoter analysis in between, without any attempt to correlate with expression of the transcription factors identified as binders in those cancers/ cancer cells, does not offer much insight.
4- Lines 331-334 suggest that the correlation between SMO and GLi2 expression levels could explain increased tumorigenesis by forced GLI1 or GLI2 expression. This does not have any rationale since activation of GLI2 and induction of GLI1 are the consequence of SMO activation, not the other way around. Inhibition of GLI1/2 or silencing is sufficient to prevent tumorigenesis in those tissues.
5- Line 337-343 are too speculative. Ptc1+/- animals have higher bone density because they have partial activation of SMO. Nothing indicates that this is due to reduced GLI3 levels. In addition, GLI3 mutations cause mostly congenital patterning syndromes, unlike PTCH1 mutations that are more related to cancer formation.
6- there are lots of grammar/style problems that indicate a need for thorough proofreading of the manuscript. For example "They reported that PTCH1 deficient (PTCH1 +/-) cells showed increase adult bone mass in patients with nevoid basal cell carcinoma syndrome".
7- I found surprising the the promoter analysis was not extended to perform luciferase reporter assays co-transfecting the TFs identified, or in cells with knockdown of the same TFs.
8- The analysis of the luciferase reporters in the three different cell lines is inadequate. First, it is not possible to compare the promoter strength among the different cell types. Second, the slight increase of the luciferase activity after the first deletion (-959 to -500 bp) of the 5' region is not observed in 2/3 cell lines, suggesting that it is due to experimental variation. Third, addition of reporter assays with small mutations in the SP1 and AP1 binding sites are necessary to properly determine their role in the regulation of SMO expression.
Author Response
Thank you very much for your comments. Please see our responses in the attachment.

Reviewer 3 Report
The authors find no correlation among the different HH pathway genes when their levels are studied in 33 different cell lines. Then they find three CpG rich islands that may work as promoters, one is located upstream of transcription initiation site (CpG1) the second is in the 5’ UTR and the third one after the ATG. They demonstrate that the CpG1 contains numerous TF binding sites specially SP1 sites. They clone CpG1 region in a luciferase vector and find that it has promoter activity in three different cell lines. Then they measure the level of methylation of the 3 CpG regions in the different cell lines and correlate it with the level of Smo expression, finding that in the cell lines where the CpG islands are highly methylated the levels of Smo expression are very low, in fact no detectable. On the other hand, they observe that in other lines where there is no methylation in these CpG inslands, the levels of Smo expression are high, although there are also several cell lines with no methylation and very low levels of Smo expression. Finally, they can increase Smo expression by incubating MCF7 cell line (this cell line had no detectable levels of Smo) with 5-aza-dC (prevents methylation).
In my opinion, the findings are interesting enough although I have a couple concerns.
1- Although the paper is well written, I really miss statistical analysis of the results, or at least the standard deviation of measurements (I am assuming that they have been performed all the experiments more than once). Especially in the results shown in Table 3.
2- In the luciferase assays shown in Fig3B, the highest promoter activity is obtained in a cell line (MCF7) that has very low expression levels of Smo and the promoter highly methylated. Whereas the other two cell lines (PC3,BT549) both express high levels of Smo and have no methylation in the promoter. These two results are not coherent; the authors should at least try to offer an explanation.
Author Response
Response to comments of Reviewer #3
Comment 1: Although the paper is well written, I really miss statistical analysis of the results, or at least the standard deviation of measurements (I am assuming that they have been performed all the experiments more than once). Especially in the results shown in Table 3.
Answer: We thank your positive comments. The real-time quantitative MSP in Table 3 was performed in two replicates of each sample. We think it’s not appropriate to calculate a standard deviation. To address your statistical concern, we have added standard deviations to Table 1 which was performed in triplicate.
Comment 2: In the luciferase assays shown in Fig3B, the highest promoter activity is obtained in a cell line (MCF7) that has very low expression levels of Smo and the promoter highly methylated. Whereas the other two cell lines (PC3,BT549) both express high levels of Smo and have no methylation in the promoter. These two results are not coherent; the authors should at least try to offer an explanation.
Answer: This is same concern was raised by reviewer #1. The MCF7 cell line has strong expression of key transcription factors, including SP1, AP2a and CREB, and it is efficiently transfected with the unmethylated luciferase reporter constructs. The undetected SMO in MCF7 cell is likely due to hypermethylation in the promoter region of SMO, and this does not affect luciferase activity.
Reviewer 4 Report
Lou et al, presented data to show molecular regulation of smoothened via promoter regulation through extensive studies in over 30 cell lines. While the conclusion was supported by cultured cells, additional in vivo information may be more helpful. The in vivo information can be obtained with public database on ATAC results from tumors or developmental embryos. In light of the significant variation on hedgehog signaling between in vitro studies and in vivo data, in vivo information will help understand how SMO is regulated.
Author Response
Response to comments of Reviewer #4
Comment 1: While the conclusion was supported by cultured cells, additional in vivo information may be more helpful. The in vivo information can be obtained with public database on ATAC results from tumors or developmental embryos. In light of the significant variation on hedgehog signaling between in vitro studies and in vivo data, in vivo information will help understand how SMO is regulated.
Answer: Thank you for your suggestion. We fully agree that in vivo information is critical to help us to understand the regulation of SMO genes. In fact, 5’Aza-C cannot fully restore SMO expression in MCF7 cells suggesting that other epigenetic mechanisms might be involved.
We now cite data using a ChIP-sequencing approach, showing that specific histone mark Histone 3 Lysine 4 Acetylation (H3K4Ac) peaks are located in the proximal promoter of the SMO and GLI1 genes, demonstrated the expression of these genes was regulated by the removal of H3K4Ac mediated by Histone Deacetylase 3 (HDAC3).
Line 322: “By using a ChIP-sequencing approach, specific histone mark Histone 3 Lysine 4 Acetylation (H3K4Ac) peaks have been confirmed in the proximal promoter of the SMO and GLI1 genes, demonstrated the expression of these genes was regulated by the removal of H3K4Ac mediated by Histone Deacetylase 3 (HDAC3) [39].”
Round 2
Reviewer 1 Report
The inclusion of more comprehensive expression data from public databases is appreciated. In future studies, it would be highly interesting to test hypomethylating agents for their impact on SMO regulation, which would have been nice to see in the present study.
Reviewer 2 Report
N/A